# Genetic Diagnosis of Retinoblastoma Using Aqueous Humour—Findings from an Extended Cohort

**DOI:** 10.3390/cancers16081565

**Published:** 2024-04-19

**Authors:** Amy Gerrish, Chipo Mashayamombe-Wolfgarten, Edward Stone, Claudia Román-Montañana, Joseph Abbott, Helen Jenkinson, Gerard Millen, Sam Gurney, Maureen McCalla, Sarah-Jane Staveley, Anu Kainth, Maria Kirk, Claire Bowen, Susan Cavanagh, Sancha Bunstone, Megan Carney, Ajay Mohite, Samuel Clokie, M. Ashwin Reddy, Alison Foster, Stephanie Allen, Manoj Parulekar, Trevor Cole

**Affiliations:** 1West Midlands Regional Genetics Service, Birmingham Women’s Hospital, Birmingham Women’s and Children’s NHS Foundation Trust, Birmingham B15 2TG, UKedwardstone@nhs.net (E.S.); c.roman-montanana@nhs.net (C.R.-M.); s.clokie@nhs.net (S.C.); stephanie.allen13@nhs.net (S.A.);; 2North West Genomic Laboratory Hub (Manchester), St Mary’s Hospital, Manchester University NHS Foundation Trust, Manchester M13 9WL, UK; sancha.bunstone1@nhs.net (S.B.); megan.carney@nhs.net (M.C.); 3Birmingham Children’s Hospital Eye Department, Birmingham Women’s and Children’s NHS Foundation Trust, Birmingham B4 6NH, UK; 4Birmingham Children’s Hospital Histopathology Department, Birmingham Women’s and Children’s NHS Foundation Trust, Birmingham B4 6NH, UKsusan.cavanagh@nhs.net (S.C.); 5Retinoblastoma Unit, Royal London Hospital, Barts Health NHS Trust, London E1 1BB, UK

**Keywords:** retinoblastoma, aqueous humour, RB1 gene, cell-free DNA, diagnosis, liquid biopsy, targeted next-generation sequencing

## Abstract

**Simple Summary:**

Identifying the genetic cause of a retinoblastoma tumour can help doctors determine any future cancer risk for the patient or their family. Somatic testing has historically used DNA from the tumour, after the eye has been removed. We and others have previously shown cell-free DNA (cfDNA) from the tumour is present in the eye fluid of retinoblastoma patients. In this study we tested eye fluid from 68 patients, collected at different points in their treatment. Measurable levels of cfDNA were found in all 11 samples of eye fluid taken from patients who had received less than three cycles of chemotherapy, and 95% (21/22) of causal genetic variants were identified. Eye fluid collected later in the treatment had 150 times less cfDNA and only 46% of genetic variants (25/54) could be detected. Eye fluid sampling early in treatment is therefore likely to be required for successful somatic testing in retinoblastoma patients undergoing eye conservation treatment.

**Abstract:**

The identification of somatic *RB1* variation is crucial to confirm the heritability of retinoblastoma. We and others have previously shown that, when tumour DNA is unavailable, cell-free DNA (cfDNA) derived from aqueous humour (AH) can be used to identify somatic *RB1* pathogenic variation. Here we report *RB1* pathogenic variant detection, as well as cfDNA concentration in an extended cohort of 75 AH samples from 68 patients. We show cfDNA concentration is highly variable and significantly correlated with the collection point of the AH. Cell-free DNA concentrations above 5 pg/µL enabled the detection of 93% of known or expected *RB1* pathogenic variants. In AH samples collected during intravitreal chemotherapy treatment (Tx), the yield of cfDNA above 5 pg/µL and subsequent variant detection was low (≤46%). However, AH collected by an anterior chamber tap after one to three cycles of primary chemotherapy (Dx1+) enabled the detection of 75% of expected pathogenic variants. Further limiting our analysis to Dx1+ samples taken after ≤2 cycles (Dx ≤ 2) provided measurable levels of cfDNA in all cases, and a subsequent variant detection rate of 95%. Early AH sampling is therefore likely to be important in maximising cfDNA concentration and the subsequent detection of somatic *RB1* pathogenic variants in retinoblastoma patients undergoing conservative treatment.

## 1. Introduction

Retinoblastoma is a childhood intraocular cancer. Over 99% of retinoblastomas are due to the loss of function of the *RB1* tumour suppressor gene [1], caused by single nucleotide variants (SNVs), copy number variants (CNVs), loss of heterozygosity (LOH), or hypermethylation of the promoter [2,3]. The bi-allelic inactivation of *RB1* can either present in the form of two somatic mutations or via an initial germline mutation followed by a subsequent somatic hit [4]. The remaining 1% of cases are caused by amplification of the *MYCN* gene in the presence of two functional copies of *RB1* [5].

Identifying the genetic cause of a retinoblastoma is critical for both the patient and their extended family. The presence of a germline *RB1* mutation conveys a diagnosis of heritable retinoblastoma, which has several additional risks beyond those of the non-heritable (somatic) form, including the development of bilateral disease, as well as non-ocular second malignancies later in life [6]. Siblings of germline carriers are also at an increased risk of inheriting the *RB1* gene mutation, from either a heterozygous or a mosaic parental carrier [1]. The detection of a germline *RB1* variant is therefore important in order to establish a patient’s additional cancer risk, as well as the screening requirements of family members. Conversely, a diagnosis of non-heritable retinoblastoma can allow patients and family members to avoid unnecessary examinations and associated anaesthesia. Increasing evidence suggests recurrent exposure to general anaesthesia at a young age may predispose to neurodevelopmental disorders or delay [7,8].

The majority of heritable retinoblastoma cases can be identified through screening of the *RB1* gene using a peripheral blood sample [1]. However, the genetic analysis of a retinoblastoma tumour can be vital for confirming heritability status. Verification of non-heritable retinoblastoma in patients with unilateral disease requires either the detection of both somatic *RB1* pathogenic variants and the exclusion of those variants in a peripheral blood sample [9] or, in very rare cases, the identification of somatic *MYCN* amplification [5]. The characterization of *RB1* causal variants in the tumour can also aid in the detection of low-level germline mosaicism [9], due to the increased sensitivity of targeted variant analysis compared to gene screening, when the pathogenic variants are unknown.

Due to the risk of extra-ocular dissemination from a tumour biopsy [10,11], the analysis of tumour DNA has historically only been possible following enucleation of an affected eye. However, several publications have now shown that the analysis of cell-free DNA (cfDNA) derived from the aqueous humour (AH) of retinoblastoma patients can be used to detect somatic variation (reviewed in [12]), including *RB1* pathogenic variants [13,14,15,16,17,18,19,20,21,22]. The collection of AH is now common in many centres, using methods proposed as part of an enhanced intravitreal chemotherapy (IViC) injection protocol [23], where extraocular dissemination of the tumour has been found to be highly unlikely [24]. IViC is routinely used for patients with disease within the vitreous, which is less responsive to primary methods of chemotherapy administration such as intravenous (IVC) or intra-arterial (IAC) chemotherapy.

In our proof-of-principle publication, we performed targeted capture-based next generation sequencing (NGS) on AH-derived cfDNA to identify causal *RB1* mutations [13]. Analysis of AH from 10 patients who had undergone an eye enucleation identified all *RB1* pathogenic variants characterised by clinical testing of the paired tumour DNA samples. In addition, we were able to detect previously unknown somatic *RB1* pathogenic variants in two retinoblastoma patients undergoing IViC treatment. The detection of somatic *RB1* pathogenic variants in AH-derived cfDNA has subsequently been replicated by a number of groups [16,17,19,20]. However, patient numbers in all studies have been limited, particularly for AH samples collected from retained eyes, the patient cohort for whom AH-cfDNA testing has the greatest clinical value.

We now report *RB1* pathogenic variant detection, plus cfDNA quantification, in AH-derived cfDNA within an extended cohort of 75 samples collected from 68 patients. This includes 34 AH samples from enucleated eyes and 41 from eyes undergoing conservative treatment. Samples were collected after either primary (PE) or secondary (SE) enucleation, by anterior chamber tap, following one or more cycles of primary chemotherapy, which consisted of IVC or IAC chemotherapy (Dx1+), or during IViC treatment (Tx).

In this study we show that our targeted NGS assay is able to detect 93% of known or expected *RB1* variants in AH samples with measurable (≥5 pg/µL) cfDNA levels. However, we find that cfDNA concentrations across AH samples are highly variable, and significantly lower in samples taken from Tx eyes, which has a marked impact on *RB1* pathogenic variant detection. Nonetheless, AH collected by anterior chamber tap after ≤2 cycles of chemotherapy (Dx ≤ 2) provides measurable cfDNA in all cases, and a subsequent *RB1* variant detection rate of 95%.

## 2. Materials and Methods

### 2.1. Patients and Sample Collection

After obtaining ethical approval (REC reference 16/EE/0528), aqueous humour samples were collected from 68 patients (71 eyes) attending the Birmingham Children’s Hospital for treatment of retinoblastoma between November 2013 and August 2022. Informed consent for genetic diagnostic testing for retinoblastoma was obtained from all subjects involved in the study.

Approximately 100μL of AH fluid was taken via clear corneal paracentesis with a tunneled 32-gauge needle within three clinical scenarios:From primary (PE) or secondary (SE) enucleated eyes prior to opening the eye for tumour dissection;During IViC via the standard clinical protocol of AH aspiration prior to chemotherapy injection (Tx);By diagnostic anterior chamber tap from retained eyes that had received a minimum of one cycle of intravenous (IVC) or intra-arterial (IAC) chemotherapy (Dx1+).

During scenarios 2 and 3, a clear corneal incision was performed with full sterile precautions and followed by triple freeze-thaw cryotherapy to the needle track.

Venous blood was collected from all patients at diagnosis. Tumour tissue was collected from 24 out of the 34 eyes that had undergone enucleation.

### 2.2. Sample Processing and Analysis

Cell-free DNA was extracted from all aqueous humour samples (*n* = 75). Quantification and fragment profiling was achieved by Qubit (Thermo Fisher Scientific Inc., Waltham, MA, USA) and/or Agilent Tapestation (Agilent Technologies Inc., Santa Clara, CA, USA). Targeted NGS was subsequently performed on AH-derived cfDNA, plus paired genomic DNA (gDNA) and tumour DNA (tDNA) where available, using KAPA Target Enrichment technology (Roche Diagnostics, Indianapolis, IN, USA) and MiSeq sequencing (Illumina Inc., San Diego, CA, USA). Sequencing data were processed by a custom bioinformatic pipeline (see Section A.3) to identify pathogenic *RB1* variation. Variants were manually inspected and filtered for a minimum total read depth of 30 or an alternative allele count ≥ 10. Variants that were designated by Platypus as having a variant allele frequency (VAF) > 20% that did not reach either the total or variant minimum read depths were given a sub-optimal calling (designated Δ is Appendix A) and not included in the main analysis. Sequence nomenclature used HGVS guidelines [25]. Pathogenicity was assessed according to Association of Clinical Genomic Science (ACGS) Best Practice Guidelines [26].

Regions of LOH were determined by calculating the beta allele frequency (BAF) from the BAM files for each informative SNP site across chromosome 13. BAFs from cfDNA or tDNA samples were then compared graphically by chromosome location against the paired gDNA. CNV analysis was performed by plotting the normalised depth of coverage for each probe target site across the *RB1* gene against a normalised reference set of patients with no CNVs reported. CNVs were manually called following inspection of regions with a normalised depth of coverage outside of the expected range (defined as ± two standard deviations from the mean). Any called CNVs were also checked against the BAF data for the region.

All variant analysis was performed blind to any previous clinical diagnostic results.

Results were compared to clinical diagnostic testing of gDNA (*n* = 68) and tDNA (*n* = 24) performed by the Manchester Centre for Genomic Medicine, North West NHS Genomics Laboratory Hub. The expected number of pathogenic variants was calculated on the assumption of two loss-of-function *RB1* mutations per tumour/eye. Full details of sample processing, extraction, quantification and NGS, as well as bioinformatic analysis, can be found in the Appendix B.

### 2.3. Previous Publication of Data

This cohort of 68 patients includes twelve previously published in Gerrish et al., 2019 [13]. The original data from eight AH samples collected from enucleated eyes (P14–16, P18, P20, P22, P23 and P32) have been included in this publication, as well as alternative sequencing results from four eyes (P4, P6, P61-OS and P62). In this dataset, repeat testing using an alternative technical protocol (P4 and P6) or a serial AH sample (P61-OS and P62) has been performed. *RB1* pathogenic variant detection was concordant across both datasets.

### 2.4. Statistical Analysis

For comparison of cell-free DNA concentrations across samples, concentrations determined by Agilent Tapestation or Qubit were normalised to a standard AVB elution volume of 70 μL.

The assessment of normality of data was performed using the Shapiro–Wilk Test. Comparison of cfDNA concentrations, as well as SNV allele frequencies, across collection groups, was performed using the Kruskal–Wallis H test. Comparison of cfDNA concentrations between individual collection groups, as well as IIRC Grade, was performed using the two-tailed Mann–Whitney U. Statistical analysis was performed using the Real Statistics Resource package [27]. Given *p* values have been adjusted for multiple testing.

### 2.5. Clinical Diagnostic Testing

Clinical diagnostic testing on genomic (*n* = 68) and/or tumour DNA (*n* = 24) was performed by the Manchester Centre for Genomic Medicine’s North West NHS Genomics Laboratory Hub. SNVs and insertion/deletions to 5% admixture were detected using long-range PCR followed by NGS of the *RB1* gene and promoter (LRG_517t1). CNV testing was performed using the P047 MLPA kit (MRC-Holland, Amsterdam, The Netherlands). LOH testing was performed using intragenic microsatellite markers Rbi.2 and RB1.20 (located in introns 2 and 20, respectively) plus extragenic marker D13S118 (0.98 Mb centromeric to Rbi2).

## 3. Results

### 3.1. Patient and Sample Characteristics

A total of 75 AH samples were obtained from 68 retinoblastoma patients (71 eyes). Patient and sample characteristics are summarised in Table 1. Nineteen patients had bilateral disease, and forty-nine were unilateral. Using IIRC standards [28], eyes were graded as C (*n* = 6), D (*n* = 38) or E (*n* = 27) at presentation.

Aqueous humour samples were taken from both enucleated eyes (*n* = 34) and eyes undergoing conservative treatment (*n* = 41), at four points in the patient pathway. Twenty-seven were from chemo-naive primary enucleated eyes (PE) and seven from eyes that had received chemotherapy treatment (mean 5.5 cycles, range 2–8) prior to a secondary enucleation (SE). Fourteen samples were collected by diagnostic anterior chamber tap following a minimum of one (mean 1.6, range 1–3) cycle of primary chemotherapy (Dx1+). Twenty-seven samples were obtained opportunistically via routine collection during IViC treatment (Tx), following an average of 5.7 (range 2–12) rounds of IVC, IAC or IViC chemotherapy.

Of the nineteen patients with bilateral disease, three had an AH sample collected from both eyes (P11 and P12 at PE/Tx and P61 at Tx/Tx). Four unilateral eyes (P45–48) had AH collected via both Dx1+ and Tx procedures.

None of the children who underwent anterior chamber taps at Dx1+ or AH collection at Tx had an adverse clinical consequence (minimum 12 months follow up) related to the procedure. There were no instances of lens trauma, hyphaema, intraocular infection, orbital recurrence, or metastatic disease.

### 3.2. Cell-Free DNA Concentration in Aqueous Humour

Cell free DNA concentration was quantified in all AH samples post-extraction using a combined approach of Qubit fluorescence and automated electrophoresis using the Agilent Tapestation system (see Section A.1). Concentrations were found to be highly variable, ranging from <5 pg/µL to 450 ng/µL, and differed significantly across the four collection points (Figure 1, *p =* 3.5 × 10^−6^).

Comparisons between individual groups determined that cfDNA levels within AH taken from PE eyes (median 2.07 ng/µL, range 0–450 ng/µL) were significantly higher than levels in both SE (median 0 ng/µL, range 0–0.34 ng/µL, *p =* 0.008) and Tx samples (median 0.003 ng/µL, range 0–1.19 ng/µL, *p =* 5.7 × 10^−5^). The cfDNA concentrations observed in Dx1+ samples (median 0.25 ng/µL, range 0.9 pg/µL–1.32 ng/µL) were also significantly higher than in both Tx (*p =* 0.006) and SE eyes (*p =* 0.04), although there was no significant difference between Dx1+ and PE samples. Further stratification of PE eyes by IIRC classification determined that the median level of cfDNA in IIRC grade E eyes (3.64 ng/µL) was markedly higher than that of grade D eyes (0.35 ng/µL), although this did not reach statistical significance (Appendix A).

As well as evaluating cfDNA concentration within the AH samples, we also compared the percentage of AH samples with cfDNA levels above the reported sensitivity limit (5 pg/µL) of Agilent Tapestation quantification [29] (Figure 2A). Twenty-four (89%) PE samples had cfDNA concentrations of 5 pg/µL or greater. A comparable proportion of AH samples from Dx1+ eyes (86%, 12/14) also provided a measurable level of cfDNA. This increased to 100% (median concentration 0.42 ng/µL, range 0.02–1.32 ng/µL) when analysis was limited to Dx1+ samples which had undergone ≤ 2 chemotherapy cycles (Dx ≤ 2, *n* = 11, Figure 2B). In contrast, measurable cfDNA was reduced to 37% (10/27) and 29% (2/7) in Tx and SE AH samples, respectively.

### 3.3. RB1 Pathogenic Variant Detection Using Aqueous Humour

Targeted sequencing of the *RB1* locus was performed on cfDNA obtained from all 75 AH samples. Paired gDNA was analysed in parallel for 53 of the 68 patients.

Thirty-four AH samples from enucleated eyes (PE and SE), for which *RB1* causal variants (germline *n* = 15, somatic *n* = 42) were known for all but one patient (P7), were analysed alongside paired tDNA from sixteen of the thirty-four cases. In addition, 41 AH-derived cfDNA samples were analysed from 36 patients undergoing conservative treatment (Dx1+ and Tx), where somatic pathogenic variants were unknown. Results are summarised below, with full details provided in Appendix A.

#### 3.3.1. *RB1* Pathogenic Variant Detection in AH from Enucleated Eyes

##### Primary Enucleation (PE)

Targeted sequencing of 27 AH-derived cfDNA samples collected from PE eyes identified 45/48 (94%) *RB1* pathogenic variants previously identified by clinical diagnostic testing. This included 24 SNVs, 14 regions of LOH and 7 CNVs. In addition, five previously unknown somatic *RB1* pathogenic variants were identified, two SNVs and three regions of LOH, in AH-derived cfDNA from patients for whom tDNA had not been previously tested. Therefore, the overall detection rate of *RB1* pathogenic variants in PE samples was 50/55 (91%) (Figure 2A), based on the expected presence of two *RB1* causal variants in 25/26 samples, plus three *RB1* pathogenic variants previously identified by routine clinical testing in patient P13.

Five *RB1* pathogenic variants were undetected across the 27 AH-derived cfDNA samples. Four of these variants were from samples P7 and P9, which produced reduced quality sequencing data (<10,000 total reads). The remaining undetected variant was a somatic *RB1* exon 1 amplification, previously identified by clinical testing of tDNA for patient P13. Analysis of the paired tDNA using our NGS assay also failed to detect this variant. Alternate breakpoints of two *RB1* deletions were also identified within AH-derived cfDNA, compared to the paired tDNA and/or clinical diagnostic results (P16 and P1, respectively)

##### Secondary Enucleation (SE)

Seven AH samples taken following SE were analysed. Overall, our pathogenic variant detection rate using AH-derived cfDNA from SE samples was 4/14 (29%) (Figure 2A), with two *RB1* pathogenic variants (one SNV and one region of LOH) identified in two of the seven AH samples. Three of these variants had previously been reported through clinical diagnostic testing. The remaining five samples provided insufficient reads for variant analysis (average total read count 20,380, range 900–70,000).

#### 3.3.2. *RB1* Pathogenic Variant Detection in AH from Eyes Undergoing Conservative Treatment

##### Anterior Chamber Tap (Dx1+)

Fourteen AH samples were collected within the Dx1+ group. This included thirteen unilateral patients and one bilateral patient (P44) for whom a germline variant of uncertain significance (VUS) had previously been detected by routine molecular genetic testing.

Across the 14 samples, we were able to detect 21/28 (75%) of the expected *RB1* pathogenic variants (Figure 2A). A range of *RB1* loss-of-function variants was observed, including seven SNVs, six regions of LOH and eight CNVs.

Three samples (P42–44) had poor quality sequencing (<30,000 total reads). These AH samples were all collected from eyes that had undergone three cycles of chemotherapy. Limiting our analysis to AH samples from eyes that had undergone ≤ 2 chemotherapy cycles (Dx ≤ 2, *n* = 11) increased our detection of expected *RB1* pathogenic variants to 95% (21/22, Figure 2B). Two *RB1* pathogenic variants were identified in the AH of 10/11 Dx ≤ 2 cases. These variants were not detected in the paired peripheral blood sample, indicating a somatic origin.

##### Eyes Undergoing IViC (Tx)

Twenty-seven Tx samples from 26 patients underwent targeted sequencing for *RB1* pathogenic variant detection. Nine patients were bilateral, one of which (P61) had an AH sample analysed from both eyes. The detection rate of expected *RB1* pathogenic variants in all Tx samples was 25/54 (46%) (Figure 2A). The twenty-five variants (fourteen SNVs, six regions of LOH and five CNVs) were identified within 14 of the 27 AH samples. The median total read count for these 14 samples was 0.94 M, compared to 0.08 M for the remaining 13 AH-derived cfDNA samples. Six further pathogenic *RB1* SNVs (denoted Δ in Appendix A) were detected in these 13 Tx samples, but did not meet our quality threshold of ≥30 total reads or ≥10 pathogenic allele reads.

### 3.4. Determining Tumour-Derived Fraction of Cell-Free DNA

In our proof-of-principle study [13] we assessed SNV allele frequency to estimate the proportion of tumour-derived cfDNA, also known as circulating tumour DNA, found within retinoblastoma AH samples. We repeated this analysis in this extended cohort. Across all cfDNA samples, a total of 33 somatic pathogenic *RB1* SNVs were detected. The zygosity for all variants could be determined due to the identification of a second *RB1* variant within the same AH sample. Mean heterozygote and homozygote pathogenic allele frequencies were 0.45 (range 0.28–0.56) and 0.95 (range 0.83–100), respectively, indicating that, in the majority of cases, >90% of cfDNA within AH from retinoblastoma eyes is tumour-derived. No significant difference in allele frequency was observed between AH collection groups.

### 3.5. RB1 Pathogenic Variant Detection Correlates with cfDNA Concentration

Across all AH collection groups, variant detection within AH-derived cfDNA was 66%, with 100/151 known or expected *RB1* variants identified. Samples with poor quality sequencing data (total reads < 0.25 million (M)) were found to account for 82% (42/51) of missed variants. We therefore investigated the impact of cfDNA concentration on sequencing quality and variant detection in our cohort in more detail.

As could be expected, we observed a correlation between cfDNA input, total sequencing read output and pathogenic variant detection rates (Figure 3). A threshold of ≥250 pg DNA input, equivalent to a cfDNA concentration of ≥5 pg/µL, corresponded with a median of 2.0 M reads and the detection of 93% (90/97) of known or expected *RB1* pathogenic variants. In contrast, only 33% (8/24) of the *RB1* pathogenic variants could be detected with a cfDNA input of > 0 < 250 pg, where the median read count was 0.24 M. A quantified cfDNA input of 0 pg produced an average of 0.02 M reads and the identification of just 7% (2/30) of the expected pathogenic variations. In total, 10 variants were detected across six AH samples (P10, P48, P52, P54, P58 and P66) where cfDNA input was less than 250 pg. Six of these variants had previously been identified by clinical diagnostic testing of the paired gDNA and/or tDNA. Three of these six, plus two previously unknown variants, were detected within AH-cfDNA (P48, P54 and P66) which were analysed on sequencing runs where no additional paired patient samples (cfDNA, gDNA or tDNA) were included. The highest variant detection rate (98%) was observed in samples with a cfDNA input ≥ 5 ng.

The removal of samples with cfDNA input < 250 pg increased *RB1* pathogenic variant detection to 98% for PE, 100% for SE, 88% for Dx1+ and 85% for the Tx samples.

### 3.6. Comparison of cfDNA Concentration and RB1 Pathogenic Variant Detection in Serial AH Samples

Four unilateral patients in our cohort (P45–48) had both a Dx1+ and a subsequent Tx AH sample taken. We were therefore able to directly compare both cfDNA concentration and *RB1* pathogenic variant detection in these serial samples.

The four Dx1+ samples were taken after one or two cycles of either IVC or IAC chemotherapy. All contained measurable levels of cfDNA (median 0.37 ng/µL, range 0.11–1.32 ng/µL), and the subsequent total read output (median 2.05 M, range 1.9–2.8 M) was sufficient for variant analysis. Two *RB1* pathogenic variants were identified in each sample, which included three SNVs, three regions of LOH and two *RB1* deletions.

Cell-free DNA concentrations in the four subsequent Tx AH samples were ≥90% lower than that quantified in the paired Dx1+ (median 0.01 ng/µL, range 0.5 pg/µL–0.14 ng/µL), and total read outputs were markedly reduced (median = 0.47 M, range 0.07–2.6 M). While two *RB1* pathogenic variants were detected in 3/4 patients, combined allele read counts of pathogenic SNVs decreased 10-fold (mean = 38.5) compared to those observed in the Dx1+ samples (mean = 382). Furthermore, SNV c.1654C > T, detected in P48, only just reached our criteria for a reportable finding of 10 pathogenic allele counts in the Tx sample. No *RB1* pathogenic variants were detected in the Tx sample from patient P47. This AH sample was taken during the patient’s first IViC treatment following three cycles of IVC.

## 4. Discussion

The detection of somatic *RB1* pathogenic variation is critical for confirming a diagnosis of non-heritable retinoblastoma. It can also aid in the identification of low-level germline mosaicism. With improved outcomes from non-surgical treatments, enucleation rates and therefore the availability of tumour DNA, are in decline [30] and an alternative option is required to enable somatic testing. We and others have previously shown that the analysis of cell-free DNA from aqueous humour can be used as a surrogate for tumour DNA for the detection of somatic *RB1* mutations [13,14,15,16,17,18,19,20,21,22], although patient numbers, in all publications, have been relatively small. Within this study we have assessed *RB1* pathogenic variant detection in an extended cohort of 75 aqueous humour samples from 68 retinoblastoma patients. Samples were collected at one of four points in the patient treatment pathway; either before any (PE) or limited (Dx1+) chemotherapy, or during (Tx) or following failed (SE) eye salvage therapy.

Targeted NGS analysis of AH samples collected from treatment-naive primary enucleated eyes (PE) determined that our capture-based NGS assay could detect 94% of *RB1* pathogenic variants previously identified by clinical testing of the paired genomic and/or tumour DNA. In retained eyes, we were also able to detect 75% of expected *RB1* pathogenic variants when aqueous humour was collected after 1–3 cycles of chemotherapy (Dx1+). This increased to 95% of variants, when samples were limited to those taken after 1–2 treatments (Dx ≤ 2). Two *RB1* pathogenic variants were identified in the AH of 10/11 Dx ≤ 2 cases. Assessment of these variants in the paired peripheral blood samples suggested a somatic origin and a diagnosis of non-heritable retinoblastoma. In contrast, variant detection in aqueous humour samples taken opportunistically during routine IViC treatment (Tx) was markedly reduced, with only 46% of expected variants identified. Our data therefore suggest that collection of aqueous humour samples early in the treatment pathway will be required to produce a cost-effective somatic testing service with sufficient benefit to the patient. A further advantage of investigative anterior chamber taps over samples taken during IViC (Tx) is that this enables genetic testing for any case undergoing conservative treatment, not merely those who require IViC. In addition to unilateral patients, this could include bilateral cases with a germline variant of uncertain significance (VUS) or undetected low-level germline mosaicism. An aqueous chamber tap has also recently been shown to provide a conclusive diagnosis in an instance of atypical ophthalmic presentation [17].

The limited variant detection rate in Tx samples could be mainly attributed to a lack of cell-free DNA in these samples. Correlation of cfDNA input and variant detection across all samples, established that a cfDNA input of ≥250 pg corresponded with the detection of 93% of known or expected *RB1* pathogenic variants by our assay. Only a third of Tx samples provided this level of cfDNA. Below this input threshold, variant detection was <35%, reducing to less than 10% when no cfDNA was detected.

In our previous study of 12 AH samples from retinoblastoma patients, we observed a wide variation in cfDNA concentration, with markedly lower readings in the two samples which had undergone IViC [13]. This finding has been confirmed in this extended cohort, where the median cfDNA concentration in eyes that had undergone an average of more than five cycles of chemotherapy treatment (Tx, SE) were 700 times lower than that seen in treatment naive eyes and 150 times lower than within eyes that had undergone 1–2 cycles of IVC or IAC. A recent publication, which assessed a variety of analytes in AH from retinoblastoma patients [31], further corroborates our finding. In this study dsDNA levels were found to be significantly higher in AH taken at diagnosis (Dx), which included samples from primary enucleated eyes as well as anterior chamber taps, than AH taken during or after treatment.

Tx samples in this study were collected opportunistically during routine intravitreal injection. In a clinical context, IViC treatment is often indicated for persistent vitreous disease or recurrence, following the administration of an initial course of IVC or IAC. Tumour volume is usually lower in these eyes than in those that have undergone just 1–2 cycles of chemotherapy. In turn, both Tx and Dx1+ eyes have significantly reduced tumour volume compared to primary enucleated eyes. In this cohort, the observed pathogenic allele frequency corroborates our previous finding [13] that >80% of cfDNA detected in aqueous humour is tumour derived. It therefore follows that a reduction in tumour mass via eye salvage therapy would be highly likely to have significant impact on cfDNA concentration in aqueous humour. This theory is further supported by the observation by Im et al. of significantly higher cfDNA concentrations in AH taken from IIRC grade D/E eyes compared to grade B/C eyes [31]. We also observe a marked increase in the average cfDNA concentration within grade E primary enucleated eyes compared to grade D, although the difference did not reach statistical significance.

If the reduction of tumour volume in treated eyes does impact cfDNA levels, it leads to the question whether investigative anterior chamber taps should be taken at diagnosis (Dx) from treatment naive eyes, when cfDNA concentrations would be at their highest. While some groups are taking AH samples at diagnosis [15,17,21,31,32], we have limited our anterior chamber taps to post one cycle of IAC or IVC treatment. We have taken this approach to minimize the potentially increased risk of extra-ocular spread, although we are not aware of any cases that support this theory, as well as negate any prospective ethical barriers of gaining consent for an anterior chamber tap at a patient’s initial diagnostic examination. While our cohort of Dx1+ patients is relatively small, our findings provide preliminary guidance that collection of AH following up to two chemotherapy cycles can provide sufficient cfDNA for a genetic diagnosis in the majority of cases. However, we welcome further research addressing whether taps at diagnosis (Dx) or following initial preliminary treatment (Dx1+), as described here, is the optimum approach.

Our data suggests that AH collection after three or more cycles of chemotherapy, does not routinely provide sufficient cfDNA for diagnostic testing. This does not mean AH samples collected opportunistically during IViC are of no clinical value. Capture based NGS systems, like the one described in this paper, require significantly more DNA input than shallow WGS or targeted amplicon based NGS. Through shallow WGS, Tx samples have previously been used to detect recurrent copy number variants, such as the amplification of chromosome 6p, the presence of which has been shown to be significantly associated with treatment outcome [33]. Moreover, AH samples taken during treatment might be useful for the monitoring, or confirmation if required, of previously identified pathogenic *RB1* variants [15]. In this clinical scenario, the detection of variants at a sub-optimal level, which we identified in several Tx samples, would be of increased clinical value. In the future, the identification of minimal DNA input thresholds, such as 250 pg ascertained in this analysis, could provide a guidance to the best test option for a given sample. In this situation, accurate quantification of cfDNA at or below 5 pg/µL, using minimal sample input, such as that conferred by the Agilent Tapestation system, would be essential.

There are several limitations with our study. Firstly, given the retrospective nature of this report, spanning almost a decade, two extraction methods and two capture methods have been used, following the discontinuation of reagents and/or the release of improved kits. However, it should be noted that all Dx1+ samples and thirteen Tx samples, which includes the four patients with a paired Dx1+ sample, have been analysed using our most recent assay protocol.

In addition, a minority of known causal variants remained undetected where DNA input and sequencing read output should have been adequate. For known variants, this was limited to our CNV analysis and included the failure to detect a somatic single exon deletion in the AH-derived cfDNA and tDNA of one patient, as well as a germline mosaic exonic deletion in the gDNA of a second individual. We also identified alternate breakpoints of a *RB1* deletion within the AH derived cfDNA of two further cases. The detection of structural variants, particularly small CNVs, is challenging for many targeted NGS-based assays [34] and, at present, CNV size detection thresholds are likely to be necessary for any retinoblastoma diagnostic service using AH-derived cfDNA. As an additional measure, we plan to confirm the presence or absence of variants within the peripheral blood by an alternative test. This approach is of particular importance for confirming low-level germline mosaicism. In the future, expanded control datasets as well as further technical or bioinformatics improvements, such as the addition of unique molecular identifiers (UMIs), may benefit the detection of all pathogenic variation, including CNVs.

While our assay targets include the *MYCN* gene, we did not have a patient in this cohort with a known *MYCN* amplification. As such we were unable to assess our assay’s ability to detect this rare event. In addition, our test does not detect *RB1* promoter hypermethylation, which accounts for approximatively 10% of somatic variants [2,3]. A recent publication has reported the detection of methylation signatures in AH from retinoblastoma patients [35], suggesting that, in cases where initial somatic *RB1* screening fails to identify two pathogenic variants, further methylation testing may be possible. However, AH sample volume is likely to be a limiting factor here. In order to enhance the efficacy of this type of combined approach, as well as any diagnostic or prognostic testing using ocular fluid, a critical next step will be to determine both the technical and clinical factors that affect cfDNA levels in the aqueous humour. In particular, the correlation of tumour volume with AH-derived cfDNA concentration, in combination with a DNA input threshold for each possible genetic testing option, would allow clinicians to determine the optimal AH sampling and testing strategy that would provide the greatest clinical benefit to the patient.

## 5. Conclusions

Analysis of tumour DNA is the gold standard for the identification of *RB1* somatic variation in retinoblastoma, with diagnostic yield > 95%. However, there is no current test available for patients undergoing conservative treatment, the numbers of which are increasing. We have shown that targeted NGS of aqueous humour-derived cell-free DNA, where DNA input is ≥250 pg, provides a *RB1* pathogenic variant detection rate of 93%. While the majority of AH samples taken during intravitreal chemotherapy do not reach this input threshold, cfDNA concentration can be increased via an anterior chamber tap after 1–2 cycles of IVC or IAC chemotherapy, resulting in the detection of 95% of expected *RB1* pathogenic variants. Our findings therefore provide guidance on the optimal AH collection time point and genetic testing strategy for the successful analysis of somatic *RB1* variation in retinoblastoma patients undergoing conservative treatment.

## Figures and Tables

**Figure 1 cancers-16-01565-f001:**
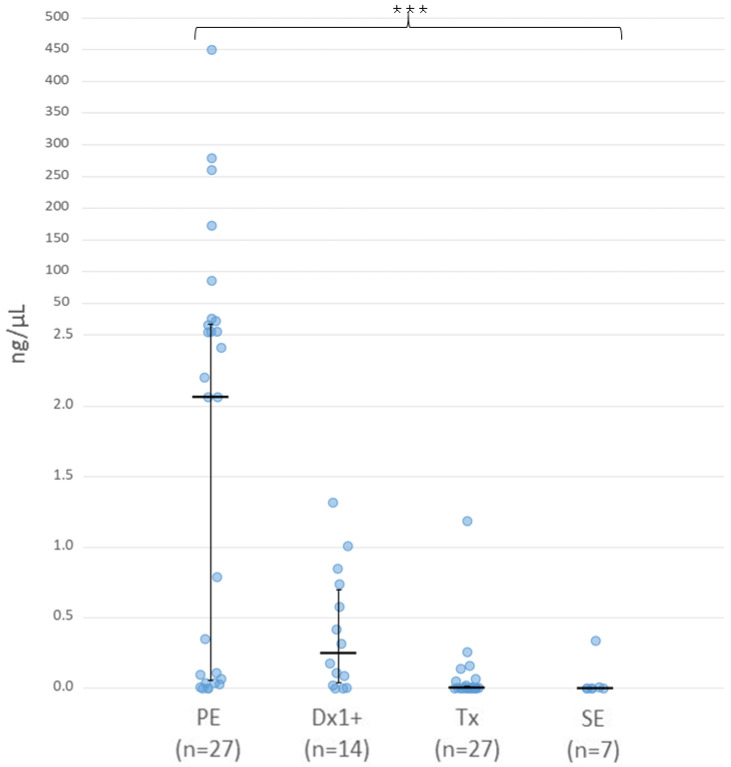
Cell-free DNA concentration of individual aqueous humour samples stratified by AH collection. Scale has been adjusted to allow the concise display of all samples. Median concentrations (black bar) with interquartile ranges (IQRs) are shown. Cell-free DNA concentrations were found to be significantly variable across the four collection groups (*p =* 3.5 × 10^−6^, *** *p* < 0.001).

**Figure 2 cancers-16-01565-f002:**
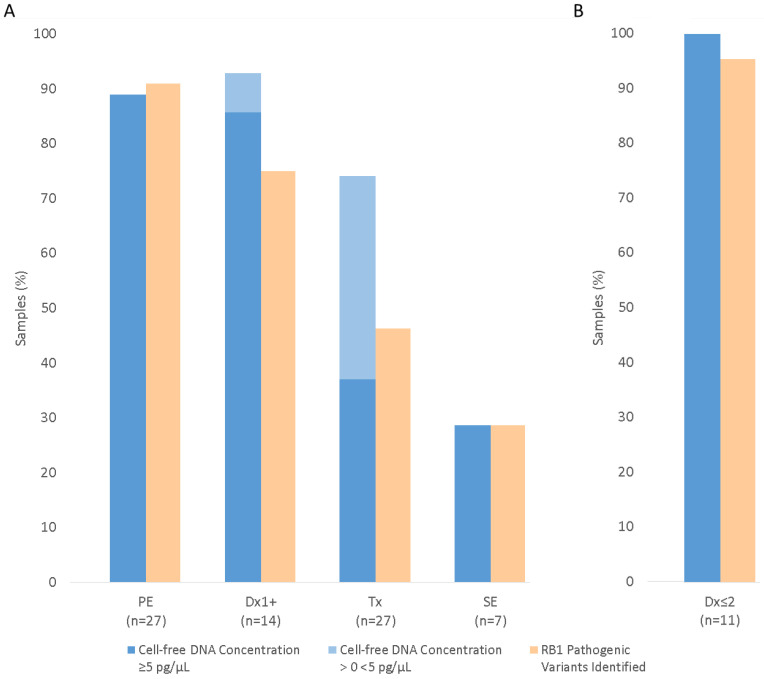
Cell-free DNA levels (blue) and *RB1* pathogenic variant detection (orange) within aqueous humour samples stratified by (**A**) AH collection, or (**B**) limited to Dx ≤ 2 samples. The percentages of samples with measurable cfDNA (≥5 pg/µL) are shown in dark blue. The percentages of AH samples with cfDNA concentrations > 0 < 5 pg/µL (light blue) are also given. The detection of known or expected *RB1* pathogenic variants is based on two pathogenic variants per sample unless >2 variants have been identified by routine clinical testing of paired tDNA.

**Figure 3 cancers-16-01565-f003:**
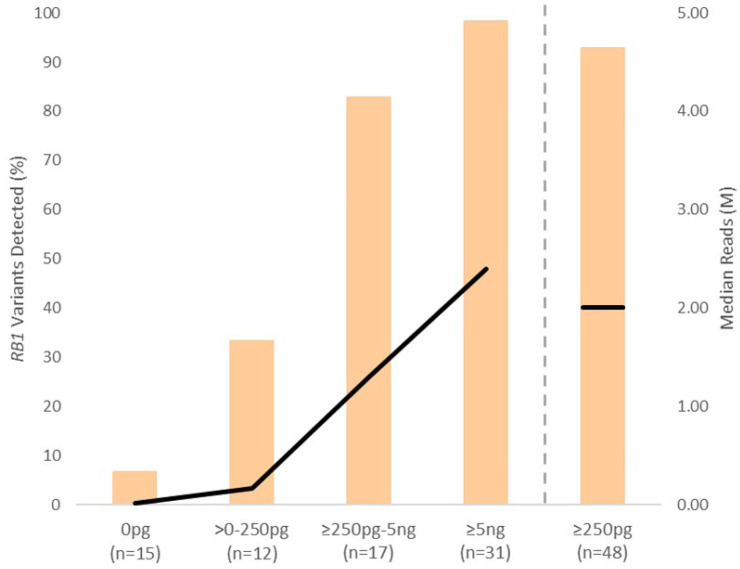
Average sequencing read output (black line) and subsequent detection rate of known or expected *RB1* pathogenic variants (orange), stratified by cell-free DNA input (pg/ng). M = Million.

**Table 1 cancers-16-01565-t001:** Characteristics of aqueous humour samples.

Sample Type	Samples (*n*)	Patients/Eyes (*n*)	Laterality U/B (*n*)	IIRC Grade C/D/E (*n*)	Chemotherapy Cycles ^1^ Mean (Range)
PE	27	27/27	21/6	0/9/18	0
Dx1+	14	14/14	13/1	5/7/2	1.6 (1–3)
Tx	27	26/27	17/9	2/19/6	5.7 (2–12)
SE	7	7/7	2/5	0/4/3	5.5 (2–8)
All	75	68/71	49/19	6/38/27	2.8 (0–12)

PE = primary enucleation; Dx1+ = diagnostic anterior chamber tap; Tx = IViC treatment; SE = secondary enucleation; U = unilateral; B = bilateral. ^1^ Chemotherapy cycles include intravenous chemotherapy (IVC), intra-arterial chemotherapy (IAC), and/or IViC treatments. A full breakdown of chemotherapy treatments is given in Appendix A.

## Data Availability

The original contributions presented in the study are included in the Appendix A; further inquiries can be directed to the corresponding author.

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
