# Peer review of "Genetic Diagnosis of Retinoblastoma Using Aqueous Humour—Findings from an Extended Cohort"

_cancers, 2024, doi:10.3390/cancers16081565_

Round 1
Reviewer 1 Report
Comments and Suggestions for Authors
This manuscript provides a compelling argument and substantial data regarding detecting somatic RB1 pathogenic variation in retinoblastoma patients. The paper's findings are robust, supported by data from a large cohort of 75 aqueous humour samples from 68 patients. The relations between cfDNA concentration and AH collection time point/treatment status are particularly important, as they help to optimize cfDNA sampling for effective RB1 variant detection. Also of note is the finding that cDNA concentrations above 5pg/ul could detect 93% of known or expected RB1 pathogenic variants as this provides a guideline for somatic RB1 variation detection in AH.
In summary, this manuscript significantly contributes to our understanding of cfDNA analysis in retinoblastoma, offering valuable insights. The detailed methodology, coupled with transparent and careful data interpretation, solidifies the research's strength, which will certainly impact genetic testing in retinoblastoma.
Author Response
We thank the Reviewer for their review and extremely kind comments.
Reviewer 2 Report
Comments and Suggestions for Authors
The authors conducted with this analysis previous results for liquid biopsy in AH in RB. In a retrospective manner the study group has collected and investigated AH of RB patients for cfDNA and somatic alterations with the objective to improve the method as a further proof of concept. With different group for investigation the authors have demonstrated progress in this topic. The manuscript is excellent written with an to the point introduction, clear method description and result display and discuss the results excellent (including a limitation section). The work improves the knowledge on this topic.
Therefore, I recommend to accept the manuscript in this form.
Author Response
We thank the Reviewer for their review and extremely kind comments
Reviewer 3 Report
Comments and Suggestions for Authors
The manuscript deals with the important issue of the applicability of liquid biopsy in retinoblastoma as a way of confirming diagnosis and distinguishing hereditable and non-hereditable cases. They analyse anterior chamber fluid of patients treated with different modalities and they observe that the highest mutation detection rates come from eyes treated with primary enucleation and the lowest rates from eyes treated with secondary enucleation and intravitreal chemotherapy, especially after multiple rounds of chemotherapy.
The manuscript is interesting and clear in general, but it could be improved by some changes. Please find below my comments:
1. I think the methods section in the main text is too short. It should include at least the “statistical analysis” section.
2. The statistical tests used to compare groups are non-parametric. Therefore, it would be advisable to report and show the median values instead of (or in addition to) the mean, in order to get a better idea of the distribution of the data points. I am referring in particular to paragraph 3.2 (Cell-free DNA concentration in aqueous humour), paragraph 3.6 (Comparison of cfDNA concentration and RB1 pathogenic variant detection in serial AH 304 samples, lines 314-322), and Figure 1. With the data presented as it is, the difference in cfDNA concentration between the PE group and the Dx+1 group is not significant but the mean is extremely different.
3. The authors should clearly state in the text that the patients from the previous proof of principle study are part of this cohort.
a. They should also state if they found any difference between the two studies.
4. It would be advisable to use the same unit of measurement throughout the manuscript. For example, the cfDNA input is sometimes expressed in pg and sometimes in ng (in paragraph 3.5 (RB1 pathogenic variant detection correlates with cfDNA concentration), lines 290-298 and in figure 3.
5. Distinction between Dx+1 and Tx groups:
a. Group Dx+1 is often described as “minimum of one (mean 1.6, range 1-3) cycles of chemotherapy”. However, IViC treatment is a form of chemotherapy as well. This distinction should be clearer.
b. As far as I understand, the procedure for AH collection in the IViC and Dx+1 group is the same and the difference is that t is already incorporated in standard practice for IViC and it was performed in addition to the standard treatment for the Dx+1 group. This concept is not clear in a few points in the text. I would therefore suggest rephrasing the following sentences:
i. Lines 107-109: “Samples were collected either after primary (PE) or secondary (SE) enucleation, by anterior chamber tap following one or more cycles of chemotherapy (Dx1+), or during IViC treatment (Tx).”
ii. Lines 159-162: “Fourteen samples were collected by diagnostic anterior chamber tap following a minimum of one (mean 1.6, range 1-3) cycles of chemotherapy (Dx1+). Twenty-seven samples were obtained opportunistically via routine collection during IViC treatment (Tx) following an average of 5.7 (range 2-12) rounds of chemotherapy”
iii. Lines 349-351: “A further advantage of investigative anterior chamber taps over samples taken during IViC (Tx) is this enables genetic testing for any case undergoing conservative treatment not just those who require IViC”
1. I would also suggest rephrasing “is this enables” to “is that this enables”
6. Reporting of variants with low cfDNA:
a. Figure 2a: in both the PE group and the Tx group, the percentage of variants identified is higher than the percentage of samples with cfDNA >5pg/μl, and 5pg/μl is reported to be the detection sensitivity limit. Were some variants detected in samples with cfDNA below the detection sensitivity limit? The authors should discuss if these variants are reliable.
b. Line 295-297 state “A quantified cfDNA input of 0pg produced an average of 0.08M reads and the identification of just 7% of expected pathogenic variation.”: can the authors address the reliability of the variants identified with a cfDNA input of 0 pg? Could contamination be involved?
7. It may be confusing for the reader to call the groups “timepoints” because not all patients underwent all the procedures listed. It would also help clarity to mention the usual timing and criteria for intravitreal chemotherapy in the introduction: this way the statement in lines 346-349 would be clearer for the reader (“Our data therefore suggests that collection of aqueous humour samples early in the treatment pathway will be required to produce a cost-effective somatic testing service with sufficient patient benefit”).
8. Abbreviations:
a. In the abstract, the abbreviation IVIC is not explained. Since it only appears once in the abstract, it could be spelled out.
9. Intravitreal chemotherapy is sometimes spelled IVIC and some times IViC. Abbreviations should be consistent.
10. In the methods section, lines 135-136 state that tumour tissue was collected for 24 out of 34 enucleated cases, but in the results section lines 208-210 say that paired analysis of tumour material was performed in 16 out of 34 cases. Do these numbers refer to different analyses?
11. Lines 58-59 read “Over 99% of retinoblastomas are 58 due to the loss of function of the RB1 tumour suppressor gene.” However, the percentage 99% is not present in references 1 and 2.
12. Lines 77-85 (“The majority of heritable retinoblastoma cases can … due to the increased sensitivity of targeted variant analysis 84 compared to gene screening when the pathogenic variants are unknown.”): this paragraph lacks literature references.
13. Line 99 “retinoblastoma patients undergoing eye conservation IViC treatment.” I would suggest rephrasing to “retinoblastoma patients undergoing eye-sparing IViC treatment.”
14. Line 102 and 130 (“in situ eyes”): I would advise against using this expression because it is not commonly used.
15. Lines 196-197: “As well as evaluating cfDNA concentration within the AH samples, we also compared the detection of measurable cfDNA levels in AH between collection groups (Figure 2a)”: this sentence would benefit from rephrasing to better show the difference between this analysis and the data from the previous paragraph.
16. Lines 214-216: “Targeted sequencing of 27 AH-derived cfDNA samples collected from PE eyes identified 45/48 (94%) RB1 pathogenic variants previously identified by clinical diagnostic testing.” From these numbers, one could gather that in 3 cases the pathogenic variant was not known, but the previous paragraph states that these was only 1 case with unknown pathogenic variant. Could the authors explain this discrepancy?
17. Please consider replacing “less” with “lower’ in lines 314, 365, 366
18. Line 339: “In conserved eyes”: please rephrase
Line 414 “There a several limitations with our study” should be “There are several limitations in our study”
Round 2
Reviewer 3 Report
Comments and Suggestions for Authors
The authors have improved the manuscript and made it clearer. They have addressed all my previous comments well. I have a few extra comments that may help improve clarity further.
1. Paragraph 3.3.2.2 (Eyes undergoing IViC): the mext says that 25 variants were detected in Tx samples, but in the Supplementary table, I count more than 25 variants detected in the Tx samples. Could the authors clarify in the table or in the text which variants they are reporting to get to the number 25? Are the variants with low cfDNA concentration presented in the table as well? If so, it would be better to clearly distinguish them in the table.
2. Lines 650-651: “Total read counts in these 14 samples was 1.36M, compared to 0.14M in the remaining 13 AH-derived cfDNA.” Are 1.36M and 0.14 average values?
3. Lines 724-728: “In total 10 variants were detected across six AH samples where cfDNA input was less than 250pg. Six of these variants has previously been identified by clinical diagnostic testing of the paired gDNA and/or tDNA. Detection of five variants within AH-cfDNA from three patients (P48, P54 and P66) occurred on sequencing runs where no additional paired patient sample (cfDNA, gDNA or tDNA) were included.”: If I sum up the 6 variants previously identified by clinical diagnostic testing and the 5 variants without paired gDNA or tDNA, I get a total of 11. Could the authors clarify this?
4. The word “naïve” is sometimes spelled incorrectly in the text
